# On the Origin of Information Dynamics in Early Life

**DOI:** 10.3390/life15020234

**Published:** 2025-02-05

**Authors:** Robert A. Gatenby, Jill Gallaher, Hemachander Subramanian, Emma U. Hammarlund, Christopher J. Whelan

**Affiliations:** 1Cancer Biology and Evolution Program, Moffitt Cancer Center, Tampa, FL 33612, USA; robert.gatenby@moffitt.org (R.A.G.); jill.gallaher@moffitt.org (J.G.); 2Integrated Mathematical Oncology Department, Moffitt Cancer Center, Tampa, FL 33612, USA; 3National Institute of Technology, Durgapur 713209, India; hemachander@gmail.com; 4Department of Experimental Medical Science, Lund University, 221 00 Lund, Sweden; emma.hammarlund@med.lu.se; 5Metabolism and Physiology Department Moffitt Cancer Center, Tampa, FL 33612, USA

**Keywords:** origin of life, information, night/day cycles, evolution

## Abstract

We hypothesize that predictable variations in environmental conditions caused by night/day cycles created opportunities and hazards that initiated information dynamics central to life’s origin. Increased daytime temperatures accelerated key chemical reactions but also caused the separation of double-stranded polynucleotides, leading to hydrolysis, particularly of single-stranded RNA. Daytime solar UV radiation promoted the synthesis of organic molecules but caused broad damage to protocell macromolecules. We hypothesize that inter-related simultaneous adaptations to these hazards produced molecular dynamics necessary to store and use information. Self-replicating RNA heritably reduced the hydrolysis of single strands after separation during warmer daytime periods by promoting sequences that formed hairpin loops, generating precursors to transfer RNA (tRNA), and initiating tRNA-directed evolutionary dynamics. Protocell survival during daytime promoted sequences in self-replicating RNA within protocells that formed RNA–peptide hybrids capable of scavenging UV-induced free radicals or catalyzing melanin synthesis from tyrosine. The RNA–peptide hybrids are precursors to ribosomes and the triplet codes for RNA-directed protein synthesis. The protective effects of melanin production persist as melanosomes are found throughout the tree of life. Similarly, adaptations mitigating UV damage led to the replacement of Na^+^ by K^+^ as the dominant mobile cytoplasmic cation to promote diel vertical migration and selected for homochirality. We conclude that information dynamics emerged in early life through adaptations to predictably fluctuating opportunities and hazards during night/day cycles, and its legacy remains observable in extant life.

## 1. Introduction

Living organisms are highly ordered, spatially distinct, semi-open systems that maintain a stable low entropy state while far from thermodynamic equilibrium [1,2] and, unlike all other structures in nature, use information to maintain this non-equilibrium state for prolonged periods of time. Information components of living systems include heritable instructions or blueprints for macromolecules in the genome that produce sensory structures that access and process information in the environment to identify risks and opportunities, which may vary both over time and across space [3].

Here, we investigate the origin of information dynamics in living systems. Virtually all origin of life scenarios recognize the critical role of information, but most focus on the prebiotic synthesis of the molecular building blocks that encode information. This work has made impressive strides in recent decades, as summarized in reviews (e.g., an edited volume on *Prebiotic Chemistry and Life’s Origin* [4] and recent papers [5,6,7], among others). From the standpoint of evolutionary biology, this body of research addresses the *proximate* (sensu Mayr [8]) or *mechanistic* (sensu Tinbergen [9]) causal bases for the evolution of life by proposing credible prebiotic synthetic pathways that generate the macromolecular components of life. Here, we propose a plausible scenario for how and why these macromolecules could be linked to information and, therefore, allow life to emerge.

While it is intuitively clear that information utilization and transgenerational propagation are essential for living systems, the definition of information can, ironically, be a source of confusion. Here, we view information as “something that reduces uncertainty”; specifically, we propose that, in early life, it reduced uncertainty about environmental changes over time; that is, we hypothesize that intracellular information dynamics originated in protocells because they correctly predict the night/day alternating environmental states and allow the protocell to anticipate and adapt to potentially lethal effects of increased temperatures and UV radiation in the daytime.

These dynamics allow and are subject to Darwinian selection and evolution; that is, our proposal concentrates on the *ultimate* or *evolutionary* causality [8,9] of protocells transitioning to life.

We assume the initial source of information could not be environmental factors that are constant or vary stochastically. Rather, early prebiotic systems could develop a “non-cognizant intelligence” to optimize their replicative ability only if the perturbation was predictably variable. Thus, we suggest that biological information dynamics were evolutionarily selected to optimize fitness under regularly changing conditions caused by night/day transitions, which imposed different opportunities and hazards. The initial biological information was specifically about time, which was linked to varying environmental conditions that affected survival. Other, longer cyclical phenomena related to the orbit of the Earth around the sun (and the moon around the Earth), including the tides, climate, and ocean currents, also represent informational cycles that drive additional adaptations. However, we hypothesize that these dynamics could impose selection forces only after a genetic storage system was in place and remained stable over prolonged periods of time.

Here, we focus on the role of diurnal environmental variations in the evolution of primordial cellular structures and information dynamics. Changing conditions produced a classic [and perhaps first] food/safety trade-off in living systems [10]. Daytime hours with increased temperature and UV irradiation accelerated chemical reactions to produce more resources and increased diffusion and convection currents in their environment, allowing greater delivery of substrates; thus, to succeed, a protocell would need to capture scarce organic molecule resources while also avoiding the damaging effects of UV radiation. Similarly, increased daytime temperatures accelerated key chemical reactions but also caused the separation of double-stranded polynucleotides, leading to hydrolysis, particularly of single-stranded RNA. We hypothesize that a protocell would need to move in its environment in a way that is referred to as “viability-based behavior” [11], defined as “a way that simple entities can adaptively regulate their environment in response to their health, and in so doing, increase the likelihood of their survival”.

We hypothesize that these cyclical variations in environmental conditions selected for temporal information dynamics in protocells containing self-replicating polynucleotides that allowed them to anticipate critical environmental selection forces and adapt to them quickly or *a priori* to optimize fitness.

If the initial evolutionary selection for information processing capabilities in early living systems emerged in response to the night/day cycle, we anticipate these properties would be observed across extant life forms. Diurnal variations in the state of biological systems are, indeed, ubiquitous across the tree of life. In past work, we have noted that the four nucleotide components of DNA in all living systems critically share an asymmetry that allows rapid synthesis of a daughter strand of DNA or RNA from the template [12,13,14]. Because increasing temperatures during the day will cause separation of double-stranded polynucleotides, synthesis must be completed during cooler nighttime temperatures, applying strong selection for faster replication and, therefore, selective inclusion of asymmetric nucleotides.

Here, we focus on other selection forces during daytime hours as the separation of polynucleotide strands, particularly RNA, is subject to hydrolysis as well as intense UV radiation that broadly damages all the macromolecular components of protocells. We propose that these Darwinian selection forces, when applied to self-replicating RNA, favored nucleotide sequences that could reduce hydrolysis by self-bonding in hairpin loops or forming RNA–peptide hybrid molecules. The hairpin loop adaptation generates a structure observed in transfer RNA (tRNA), initiating proposed early life dynamics associated with tRNA (see [15,16]). RNA–peptide hybrids, when trapped within the protocell membrane, could promote protocell replicative success by protecting them from UV damage through scavenging free radicals or by catalyzing the synthesis of melanin from tyrosine.

We hypothesize that these dynamics persist in modern living systems as transfer tRNA, RNA-templated synthesis of proteins, ubiquitous diurnal rhythms, and melanosomes, which are evolutionarily ancient [17,18] membrane-bound organelles in which melanin is synthesized. Additionally, we propose that early living systems could limit UV damage by vertical migration facilitated by replacing intracellular Na^+^ with heavier K^+^ ions, leading to the transmembrane ion gradients ubiquitously observed in modern life. This may constitute the origins of the diel vertical migrations observed across numerous aquatic organisms today [19].

## 2. Building a Model of Information in Early Life

We assume the initial conditions of an aquatic environment during the Hadean Eon [4.3–4.0 billion years ago; Ga] or possibly the Eoarchean Era [4.0–3.6 Ga]. At this time, small continents [20] and volcanic islands had formed [21]. Putative chemical evidence of early life is preserved in sedimentary rocks from this age (3.95 Ga) [22]. The origin of life is often linked to hydrothermal vents because they offer ample chemical energy and the deep ocean niche was shielded from UV radiation and interruptive meteorite bombardment as late as ~3.5 Ga [23]. In contrast, shallow shelf surfaces of early continents indeed would be exposed to not only bombardment of meteorites but also daytime UV radiation, which was much higher than at present, owing to a more active [although less luminous] sun [24] and absence of an ozone layer [25]. Despite the assumed destructive nature of meteorite impacts, the high temperatures generated may have resulted in reactions that produced necessary feedstocks from which RNA, proteins, and lipids ultimately arose [26].

Our assumed initial conditions include the availability of lipids, polypeptides, and nucleotides that formed through condensation reactions [27]. Amphiphilic compounds (those with polar and nonpolar components) can spontaneously self-assemble into bi-molecular layers, forming closed membranous vesicles [28]. Such vesicles in an aqueous environment rich in organic molecules could entrap those organic molecules, including self-replicating polynucleotides as well as mononucleotides and amino acids. These vesicles, with their entrapped molecular components, are self-assembled protocells, which can subsequently undergo replication subject to Darwinian selection, ultimately leading to the evolution of modern life as we know it today [29].

## 3. Fulfilling Darwin’s Postulates Under Primordial Conditions

We propose that a critical requirement for the origin of life is a non-dissipative environment, such as a tidal pool or small lake, that physically constrains both self-replicating polynucleotides and protocells, as well as their essential molecular building blocks [30]. Here, nucleotides, amino acids, fatty acids, and other molecules are present in the environment from local prebiotic synthesis with sustained or periodic influx of substrate [7], but in limited quantities. This physically constrained environment requires self-replicated polynucleotides and protocells to compete for limited substrates and space, thus imposing prebiotic evolutionary dynamics. This contrasts with ocean regions around thermal vents, where molecules generated by the vent would diffuse rapidly into the oceanic environment.

We note that these conditions fulfill Darwin’s three postulates [31], which are necessary for evolution by natural selection. Self-replicating polynucleotides that can affect the phenotypic properties of the protocell provide the necessary heritable variations. The diurnal fluctuations of benefits and threats in a resource-limited, non-diffusive environment apply selection pressures that determine proliferation [32]. Those most adapted to the cycling opportunities and hazards will survive and replicate at the expense of those less capable, generating a “struggle for existence”.

## 4. Diurnal Variations in Evolutionary Selection Pressures

We next hypothesize that critical information available to early self-replicators arises through variation in selection forces during the regular night/day cycles (Figure 1). Although tides in the deep ocean niches may result in temperature and energy variations near hydrothermal vents [33], the influence of time on vent activity is an open question [34]. However, oscillations in shallow marine settings encompass sharper diurnal variations in UV light, oxidants, and temperature.

In daylight hours, intense UV exposure could drive prebiotic chemistry, including the formation of iron–sulfur clusters [35]. Noting that solar radiation is the paramount source of energy on Earth and can lead to the synthesis of organic molecules from nonorganic building blocks, Green et al. [36] identified photochemical pathways or schemes relevant to prebiotic molecules and reactions (such as condensation of biomolecules). A growing body of literature reports that UV irradiation can induce reactions that produce prebiotic precursors to the chemical building blocks of life [36,37] under plausible environmental conditions of early Earth. As stated by Nunes Palmeira et al. [38] “The idea that metabolism emerged from a geochemical protometabolism therefore looks increasingly persuasive”.

However, opportunities for protolife generated by daytime production of the substrate from prebiotic chemistry are countered by hazards, generating a food and safety tradeoff [39]. Green et al. [36] emphasize that even as UV radiation provides energy for the formation of chemical bonds and the synthesis of biomolecules, it can break down organic molecules via photolysis. Mulkidjanian and Junge [40,41] proposed that modern photosynthetic reaction centers evolved from earlier molecular “technologies” that evolved to protect “primordial” cells from UV damage. This destructive nature of UV radiation is the driving force of numerous protective strategies in modern organisms [42].

In addition, diurnal variations produce daily temperature changes. These regular temperature cycles, similar to Polymerase Chain Reactions (PCRs), could promote self-replicating polynucleotide strands [14]. However, warmer daytime conditions may “melt” hydrogen bonds, causing the separation of double-stranded polynucleotides. Subramanian et al. [13,14] demonstrated that this selected monomers that permitted extremely fast synthesis of complementary strands for polynucleotides to complete strand duplication prior to the onset of warmer daytime temperatures and strand separation. Molecular modeling of strand synthesis [13,14] demonstrated a strong selection for asymmetric monomers that produce increased energy in hydrogen bonds in the existing strand, thus reducing the probability of spontaneous separation while decreasing bonding energy at the developing edge of the growing strand to increase the probability of bonding with a monomer in solution.

While the melting of double-stranded polynucleotides provides a theoretical opportunity for replication, single-stranded polynucleotides, particularly RNA, will be subject to rapid hydrolysis [42]. Evolutionarily, this precludes self-replication as well as loss of any potential function of the RNA strand.

Similarly, increased daytime temperatures cause the denaturation of folded polypeptides and increase the permeability of lipid bilayers. The former yields a prediction that proteins selected in early life would remain stably folded through the range of extant environmental temperatures. Consistent with this, we note that *E. coli* has a more thermostable proteome than human cells [43], and critical proteins, such as those in respiratory chains across the tree of life, are thermally stable. Interestingly, tyrosinase, critical for the synthesis of melanin (see below), is stable between 20 and 65 C [44]. Further, increased cell membrane permeability may allow more building blocks into the protocell and thus contribute to the food part of the tradeoff.

## 5. Evolutionary Heritage of Prebiotic Information Acquisition

### 5.1. Evolution of Circadian Clocks

We propose that cycling selection forces in prebiotic life specifically favored an ability to assess and respond to information about time. As noted by Green et al. [36], early life would have evolved primitive mechanisms to take advantage of the benefits of UV radiation, while simultaneously avoiding its corrosive effect of photolysis would have promoted some mechanism to time their activities. Similarly, Baluška and Reber [45] noted that “circadian clocks are inherently cognitive in nature”, and they must have “co-evolved with the first cells to safeguard their survival”. Although it was long believed that circadian clocks within prokaryotes were restricted to cyanobacteria, the recent demonstration of a circadian clock in Bacillus subtilis [46] suggests circadian clocks may be widespread among prokaryotes, as predicted by our hypothesis.

### 5.2. The Critical Role of Simultaneous Hazards and Interacting Adaptations

We emphasize that the onset of daylight hours produces three significant changes that constitute strong selection forces on protocells: (1) UV radiation, which can broadly damage the macromolecular structures of the cell; (2) increased temperature, which can cause separation of double strands of polynucleotides; and (3) single-stranded polynucleotides, particularly RNA, are subject to rapid degradation by hydrolysis, thus effectively eliminating the possibility that long polynucleotide strands could consistently self-replicate.

We propose that these (roughly) simultaneous selection forces were critical for the development of capabilities to assess temporal information that anticipated the onset of daytime hazards and, thus, prevented hydrolysis while simultaneously producing changes that protected the protocell from UV damage; that is, the melting of double-stranded RNA due to increasing temperatures during daytime hours and the subsequent adaptations to prevent hydrolysis also produced molecular machinery to shield cellular components (e.g., membrane) from damage from UV light. We hypothesize that this is the origin of the informational dynamics still observed in the circadian changes of extant living systems [47].

Several scenarios may provide protective effects. Recent works have shown that peptides can bind to single-stranded RNA, producing a hybrid molecule that is relatively stable [48,49]. We hypothesize that the bonding energies of three nucleotides with a single amino acid in the hybrid molecule and their effects on its function and reproduction led to the triplet RNA code for amino acids. Furthermore, we note that RNA strands that form hairpin configurations, similar to the structure of modern tRNA, are relatively shielded from hydrolysis [47]. Interestingly, this mechanism of selection suggests a distinct evolutionary pathway for tRNA parallel to but in synchrony with the DNA/RNA sequences [15]. For example, Lei and Burton [16] have proposed that tRNA could directly contribute to survival and proliferation by synthesizing polyglycine as a cross-linking agent to stabilize protocells. Consistent with this model, tRNAs integrate non-canonical nucleotides in tRNA [48].

We note several potential mechanisms through which strand melting and adaptation to hydrolysis could shield the protocell from UV radiation. We assume that early self-replicating polynucleotides would be relatively short and, thus, capable of producing small peptides that can have both antioxidant and UV shielding properties. Recent work also reports that a hybrid RNA/protein molecule can synthesize melanin from tyrosine in the absence of tyrosinase [50]. Because tyrosine can be generated by prebiotic synthesis [51], it seems plausible that even very primitive early living systems could have synthesized melanin to protect from UV damage. We note that such cells would likely have been highly successful competitors, and these dynamics may be preserved in modern life as melanosomes, which are evolutionarily ancient [17] membrane-enclosed intracellular structures that are broadly involved in stress response and specialize in melanin production [52].

The melanin family of proteins is diverse and can form heterogeneous (often cross-linked) and assembled oligomers that interact with metals and can form amyloid fibrils [53]. While they are most recognized for shielding cells from UV radiation, melanin proteins have diverse cellular functions, including immune interactions, calcium homeostasis [54], and endosomal sorting [17,54]. Melanosomes are found in all living organisms, from bacteria to mammals, and were observed in early evolutionary life forms [55,56]. Melanosomes are ubiquitous across the tree of life [17]. Moreover, melanosome formation is strongly linked to intraluminal pH, which is maintained by the extrusion of Na+ via TPC2 [57,58], which may influence buoyancy as an adaptation to avoid UV light (see below).

### 5.3. Evolutionary Selection for the Transmembrane Ion Gradient

One of the characteristics demanding an explanation in the origin of life research is the relative concentrations of potassium (K^+^) and sodium (Na^+^) in the cytoplasm compared to extracellular fluids and seawater, which is often hypothesized to be the environment in which early life originated. The cell cytoplasm of all living organisms contains a high concentration of K^+^ and a low concentration of Na^+^ relative to their environment. About one-third of a cell’s energy budget is used to pump ions against a concentration gradient [59]; however, the corresponding survival benefit to justify this expenditure or resources remains unclear. As pointed out by Hansma [60], either life originated in an environment with a high concentration of potassium, or, after life’s origin, early cells evolved a preference for potassium over the more ubiquitous sodium (which occurs in high concentrations in seawater). Even if early living systems developed in a high K^+^, low Na^+^ environment, maintaining these intracellular concentrations in a seawater environment would require large investments of energy. We suggest that this strongly indicates that the high K^+^, low Na^+^ intracellular environment confers a pronounced evolutionary benefit.

Within the context of our hypothesis, we propose that any mechanism that may allow a protocell to control its depth in the thermal pond (or sea) with respect to the light/dark circadian cycle would be highly advantageous—allowing the cell to benefit from the UV -driven selective synthesis of biomolecules during daylight, while simultaneously reducing UV-driven photolysis of those biomolecules. For instance, many aquatic prokaryote species possess a gas vacuole, which confers the cell with buoyancy, and their widespread occurrence among bacteria and archaea suggests that these organelles evolved very early [61]. Before the evolution of the gas vacuole, a rather sophisticated cell organelle, we hypothesize that protocell buoyancy was controlled by altering the intracellular K^+^ and Na^+^ concentrations via channels, transporters, and pump-like membrane proteins that early cells or protocells must have developed to counter the Gibbs–Donnan effect [62]. Sephus et al. [63] suggest that a likely early (primitive) proton pump provided the active transport needed to avoid cell rupture. This primitive proton pump may have been constructed from primordial, ancestral rhodopsins that evolved over “a span of time encompassing Earth’s early history of inhabitation”. Morowitz ([64] p. 166) offered a similar perspective: ”the conversion of photon energy to a chemically useful form may have been contemporaneous with life’s origin”. We stress that, following Naranjo [62] and Sephus et al. [63], we view these as critical steps in the transition from a protocell to LUCA (Last Universal Common Ancestor), but these steps do not, in and of themselves, represent the emergence of LUCA. Indeed, various lines of evidence suggest that components of ATP-synthase predate LUCA [65,66], lending credence to the hypothesis that primordial protocells had the ability to control internal osmolarity via primitive pump mechanisms.

Assuming that early organisms or protocells contained impermeant molecules such as nucleic acids and negatively charged metabolites, the Gibbs–Donnan effect will cause cellular swelling and rupture if not countered by the extrusion of sodium by a sodium pump. The “pump-leak” model [67] proposes that a combination of Cl^−^, Na^+^, and K^+^ channels, together with Na^+^/H^+^ exchangers and a bacteriorhodopsin-like H^+^ pump, must have allowed pre-LUCA cells to resolve the Gibbs–Donnan effect. With these mechanisms in place, selective intake of K^+^ and extrusion of Na^+^, because of the greater mass of K^+^, would cause the protocell to sink a few cm in the water column, thus providing a small amount of shielding from UV radiation (Figure 2). Trapp [68] refers to critical steps like this as “a transition from chemical to biological systems”.

In the context of our hypothesis, this predicts that RNA–peptide hybrids with the capacity to transport ions across cell membranes would have been selected in early life—a prediction that can be addressed experimentally.

## 6. Discussion

Living systems, singularly in nature, use information to produce a highly ordered state that is far from thermodynamic equilibrium [1,2]. Here, we present a hypothesis that diurnal variations in environmental conditions generated the selection forces through which information became critically linked to survival and proliferation and, therefore, simultaneously gave rise to life and Darwinian evolution.

The origin of life on Earth is a topic of extensive prior research. Much of this work appropriately investigates the prebiotic chemistry that produces the molecular building blocks of extant living systems. Here, we consider the evolutionary selection forces that could plausibly select for the integration of information into the dynamics of systems that would eventually be defined as “living”. While prebiotic chemistry defines *how* the molecules necessary for information dynamics were produced, our hypothesis focuses on *why* molecularly encoded information became necessary for protocell survival; that is, using the Shannon definition of information [69], any random process contains 0 bits of information. Thus, information requires some non-random distribution of (in this case) nucleotides. Under prebiotic conditions, which permit the self-replication of polynucleotides, information is gained when a specific sequence benefits the self-replication process. Under our assumed initial conditions that include self-replicating polynucleotides entrapped within a self-replicating lipid sphere, specific sequences of polynucleotides can become promoted (i.e., persist within multiple generations of protolife) if they increase the probability of self-replication of *either the RNA or the enclosing lipid membrane.* Here, we propose that RNA sequences that enhance self-replication by reducing loss for hydrolysis can benefit self-replication of the lipid membrane by reducing damage associated with UV radiation. As a result, polynucleotide synthesis transitions from a purely random process to one in which specific sequences are repeatedly synthesized because they maximize the probability of survival and propagation of the protolife combination of RNA entrapped within a lipid membrane. These non-random sequences now carry information (calculated as bits using the standard Shannon probability equation) that benefits the nascent living system and can be propagated across generations. This fulfills Darwin’s requirement for a “mechanism of inheritance”. In effect, these dynamics simultaneously give rise to both life and evolution.

We propose that the initial source of information required a process that was neither stable nor stochastically changing, but rather, information specifically about time can be obtained only from a process that is regularly and predictably variable; thus, we propose that the information dynamics central to living systems originated in diurnal variations in environmental conditions. In particular, the daytime conditions simultaneously imposed two different opportunities and hazards: warming temperatures and high levels of UV irradiation.

These conditions would accelerate prebiotic chemistry that produced the molecules necessary for life. However, UV radiation could produce potentially fatal damage in self-reproducing systems, and the increased temperature could cause the separation of self-replicating polynucleotides. The latter exposes polynucleotides, particularly RNA, to degradation by hydrolysis and effectively limits the size of self-replicating RNA.

We note that the temporal coincidence of these processes selects for simultaneous adaptive strategies that were critically interrelated. Thus, self-replicating polynucleotides, subject to the selection pressure of strand separation during warmer daytime hours, must optimize the speed of self-replication. This will select for [12,13,14] asymmetric nucleotides that accelerate daughter strand synthesis by increasing the energy of hydrogen bonds between strands proximal to each added nucleotide (reducing the probability of separation) while decreasing the energy needed to form hydrogen bonds at the leading edge (maximizing the probability of binding by the next nucleotide in the growing daughter strand). The legacy of these dynamics persists as the nucleotides in extant living systems are all asymmetric [13,14].

In addition, self-replicating RNA can adapt to hydrolysis by producing strands that self-bond in hairpin configuration or by forming RNA–peptide hybrid molecules. The former adaptation generates precursors to tRNA and its potential roles as an (initially) independent driver of adaptation in early life. The latter can benefit the survival of the whole protocell by scavenging free radicals to limit photolysis. The latter adaptation provides a mechanism for the triplet code based on the energy of interaction of the three nucleotides in the RNA with amino acids in the peptide. Over time, this primitive transfer of information from RNA to protein can give rise to a more complex system that includes messenger RNA (mRNA) and ribosomes that combine specific RNA sequences (ribosomal RNA [rRNA]) with specialized (ribosomal) proteins. Furthermore, an RNA–peptide hybrid has been shown to catalyze the synthesis of melanin from tyrosine and may persist in modern organisms as melanosomes, which are membrane-bounded organelles that specialize in melanin production and are associated with stress responses [70]. We note that a cell-level adaptation to optimize scavenging for UV-synthesized molecules and reduction of UV-induced damage might include replacing intracellular Na+ with heavier K+ ions, thus causing the protocell to sink below the surface. This predicts the selection of RNA–peptide hybrids that can facilitate this ion transfer. Finally, the RNA–peptide hybrid anticipates the RNA-to-protein information transfer observed in all extant life.

While the origin of life scenarios are necessarily speculative, we note that our model predicts properties observed in extant life, including the presence of diurnal rhythms, diel vertical migrations across aquatic life, the configuration of tRNA, the ubiquity and broad survival-promoting activities of melanosomes, and the uniform inclusion of asymmetric nucleotide monomers in DNA and RNA. As noted above, the temperature-associated selection for hairpin loops to prevent hydrolysis of single RNA strands suggests a separate origin for tRNA and allows an initial evolutionary role independent of self-replicating DNA and RNA and RNA–peptide hybrids. Consistent with this, we note that modern tRNA uniquely includes non-canonical nucleotide monomers [15,16].

Elements of our hypothesis are testable by observing whether, e.g., simple peptides alter their configurations and interactions with other peptides or nucleotides after being exposed to cyclical changes in physical conditions such as temperature, UV, or dry/wet cycles. Furthermore, a mixture of random single-stranded RNA polynucleotides could be exposed to increased temperature and/or UV light, and the “surviving” RNA strands could be identified. We note that our hypothesis that RNA–peptide hybrids catalyze the synthesis of melanin from tyrosine is based on experimental observations. Adding self-replicating liposomes to this experiment and comparing the effects of UV light on the liposomes with and without the synthesis of melanin could be performed. Perhaps the most sophisticated (and difficult) experiment would include liposomes containing some RNA fragments and their nucleotide precursors in a culture media rich in mononucleotides and lipids (allowing liposome self-replication) that are exposed to cyclical variations in temperature and UV light.

Our hypothesized buoyancy mechanism could be tested experimentally in the laboratory with synthesized liposomes in which the internal ion concentration could be biased toward Na^+^ or K^+^. If our mechanism is correct, liposomes within which the internal concentration of K^+^ is greater than the internal concentration of Na^+^ should sink lower in an aqueous solution compared with liposomes with the relative ion concentrations reversed. Furthermore, our hypothesis predicts that membrane-associated ion pumping functions should be experimentally observable in RNA–peptide hybrids.

## 7. Conclusions

In conclusion, although the shallow marine environment of early Earth presented a limited and demanding habitat for life, it was not without opportunities. We hypothesize that the diurnal fluctuations of these limitations, demands, and opportunities are directly responsible for the information dynamics that distinguish life on Earth from all other processes in nature.

## Figures and Tables

**Figure 1 life-15-00234-f001:**
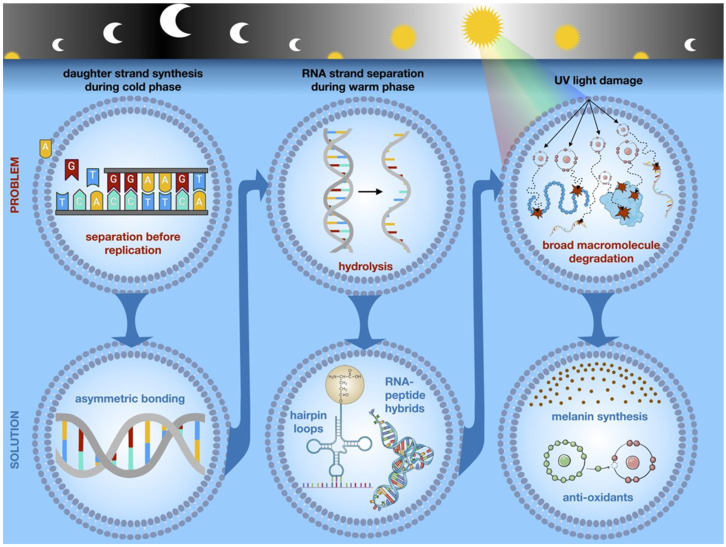
Imposed selection pressures and adaptive strategies in self-replicating RNA within a liposome. Rapid completion of strand duplication in night hours before the onset of warmer daytime conditions and strand separation selected for asymmetric nucleotides. Strand separation resulted in hydrolysis of individual nucleotide strands selecting for sequences that permitted self-binding in hairpin loops or formation of RNA–peptide hybrids. These adaptations evolved into transfer RNA (tRNA) and ribosomal RNA (rRNA), and messenger RNA (mRNA)-templated protein synthesis. UV light during daylight hours degraded all elements of the cell. However, the selection of specific RNA–peptide hybrids allowed for the scavenging of free radicals and promoted the synthesis of melanin from tyrosine, allowing protocells to survive and replicate.

**Figure 2 life-15-00234-f002:**
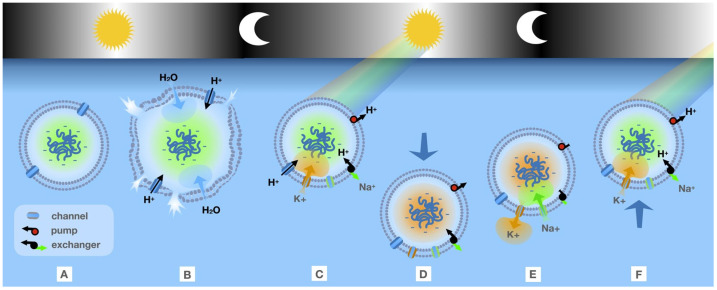
A hypothetical sequence of events that would allow protocells to reduce the damage from UV light during daytime hours. A. Large negatively charged macromolecules confined to interior create charge gradient. B. Proton entry induces osmotic imbalance leading to water entry, swelling, and bursting. C. UV-driven pump extrudes H^+^ to prevent swelling, Na^+^/H^+^ exchanger extrudes Na^+^, causing K^+^ to influx. D. Heavier K^+^ ions cause sinking. E. Pump stops during nighttime, leading to Na^+^ reentry. F. Cycle repeats on daytime, providing mechanism to shield cell from harsh UV.

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
