# Peer review of "On the Origin of Information Dynamics in Early Life"

_life, 2025, doi:10.3390/life15020234_

Round 1
Reviewer 1 Report
Comments and Suggestions for Authors
The authors present another set of hypotheses regarding life's origins, specifically, how RNA and peptides could have evolved under a set of different selection pressures, like day-night cycles.
The problem with this manuscript, although it is very well written, and it is a problem shared with all manuscripts of this ilk, is that while many interesting hypotheses are presented, not a single mention is presented about how one may go about testing any of them experimentally. The field is completely saturated with hypotheses papers -- what is needed desperately are experiments to test them.
The reviewer would recommend publication if the authors, for every hypothesis they present, suggest a way it could be tested -- even if the technology does not yet exist. How can we go about testing these hypotheses, whether the authors can suggest some very specific experiments or even just generally what could be done, that would be satisfactory.
Reviewer 2 Report
Comments and Suggestions for Authors
Review
I enjoyed reading this paper, which I found thoughtful and well-presented. Below, I have some suggestions for the authors to consider to possibly extend some of their arguments.
If RNA has a 2’-O-methyl group, it is stable to hydrolysis. Also, 5’à3’ replication becomes more plausible to evolve with this modification. Why not start with modified RNA?
See [1]
Much more might be said about the evolution of genetic coding [2-5]. This reviewer considers genetic coding to have evolved around tRNA and the tRNA anticodon. Because tRNA sequences are highly patterned (RNA repeats and inverted repeats), the pre-life tRNA sequence is known almost to the last nucleotide, giving insight into how tRNA evolved. mRNA, DNA, ribosomes, AARS (aminoacyl-tRNA synthetase) enzymes and tRNA modification enzymes are all coevolved with the code.
Evolution of the first cells might require an “aggregator” to pack interacting molecules together to promote pre-life chemical evolution. This reviewer suggests dirty polyglycine as the initial aggregator. It is a reasonable suggestion that ACCA-Gly, a GCG repeat, ribozymes and wet-dry cycles should be sufficient to synthesize polyglycine. The effects of UV light on polyglycine should be tested.
For homochirality selection, would a replicative ribozyme select chiral reagents?
Is there a reason that night-day cycling is more of a determining evolutionary factor compared to tidal and gravitational effects?
Suggest: do not use “since” to mean “because”. Use “since” to indicate the passage of time only in formal English.
1. Muller, F.; Escobar, L.; Xu, F.; Wegrzyn, E.; Nainyte, M.; Amatov, T.; Chan, C.Y.; Pichler, A.; Carell, T. A prebiotically plausible scenario of an RNA-peptide world. Nature 2022, 605, 279-284, doi:10.1038/s41586-022-04676-3.
2. Lei, L.; Burton, Z.F. The 3 31 Nucleotide Minihelix tRNA Evolution Theorem and the Origin of Life. Life (Basel) 2023, 13, doi:10.3390/life13112224.
3. Lei, L.; Burton, Z.F. "Superwobbling" and tRNA-34 Wobble and tRNA-37 Anticodon Loop Modifications in Evolution and Devolution of the Genetic Code. Life (Basel) 2022, 12, doi:10.3390/life12020252.
4. Lei, L.; Burton, Z.F. Evolution of the genetic code. Transcription 2021, 12, 28-53, doi:10.1080/21541264.2021.1927652.
5. Lei, L.; Burton, Z.F. Evolution of Life on Earth: tRNA, Aminoacyl-tRNA Synthetases and the Genetic Code. Life (Basel) 2020, 10, doi:10.3390/life10030021.
Reviewer 3 Report
Comments and Suggestions for Authors
The manuscript, "On the Origin of Information Dynamics in Early Life," by Gatenby et al speculates how diurnal environmental variations, such as day-night cycles, could have influenced the emergence of information processing and adaptive behaviors in early life forms. The authors propose that these cycles created selection pressures on primitive protocells, driving the development of molecular systems capable of responding to external stimuli.
Overall, the paper posits that diurnal cycles played a critical role in shaping the origins of life's informational complexity, driving the evolution of protocells into more advanced living systems.
The major problem with the current manuscript is that it does not explain the origins of the information to begin with. There is no solid description of the prebiotically plausible model that result in the emergence of RNAs that are capable of carrying and transferring the information. The manuscript does not states the most critical issues that need to be addressed in order to provide a comprehensive scenario on the origins of informational biopolymers: origins of complex building blocks (nucleotides), feasibility of oligomerization under prebiotic conditions, and origins on the information encoded within RNA. The evolution of informational complexity and its capacity to encode a given state of the system is not considered in the current manuscript. Additionally the regulatory mechanisms that the authors proposed for regulating the earliest information appear to be more complex than the systems that they expect to regulate. The section on evolution of homochirality does not contain any original or insightful information.
Unfortunately, I cannot recommend this submission for publication in Life.
Reviewer 4 Report
Comments and Suggestions for Authors
This paper discusses the selective effects that would act on early cells as a result of day-night cycles, and the adaptations that a cell might develop to do well in such conditions. I find the title is rather misleading – “origin of information”. In fact the paper does not discuss the origin of information at all, it presumes that there are RNAs and peptides that already have sequence information and that there are cells that are able to accurately pass information on to their descendants. The paper discusses what kinds of genes will be selected, given that genetics already exists. It does not explain how genetics originates. I would suggest changing the title to something like “Selective effects and adaptations in early cells”, or “Mechanisms by which early cells could respond to day-night cycles”.
The paper already says something similar to my previous paragraph on lines 46-48. “In contrast, we focus on the evolutionary conditions in which information becomes critically linked to survival and proliferation and, therefore, both allows and is subject to Darwinian selection.” But it seems to me that if information, heredity, and cell reproduction already exist, these are bound to be linked to survival and proliferation. Darwinian evolution will happen whatever the environmental conditions. Understanding the origin of Darwinian evolution requires understanding the molecular processes that allow sequence information and replication.
line 58 - The paper argues that prebiotic systems will “optimize their replicative ability only if the perturbation were predictably variable”. It seems to be arguing that constant conditions will not work, and randomly fluctuating conditions will not work, and only regular cycles will work. I think this is too strong. If information and heredity exist, then organisms will adapt to whatever the conditions are. It may well be true that regular day-night cycles were a key part of the environmental conditions, but this is not the same as saying that life could only evolve because there were day-night cycles.
line 103 – “selection forces, when applied to self-replicating RNA, favored nucleotide sequences that could reduce hydrolysis by self-bonding in hairpin loops or forming RNA-peptide hybrid molecules.” This seems reasonable, but it is easy to find random RNAs that form hairpin loops, and while secondary structure formation might favour stability, it might also slow down replication. There is no discussion of how replication occurred, or whether it was inhibited by too-strong secondary structure. It is easy to say that a tRNA evolved because it has a folded structure that is resistant to hydrolysis (e.g. line 206), but the hard part is to explain how a tRNA gets charged with an amino acid, and how the genetic code, translation, and ribosomes came to function. This paper seems to be ignoring all the interesting hard parts.
line 137 – I am confused by the term “non-dissipative environment”. Life itself must be dissipative. A cell has continued input, turnover, and output – a non-equilibrium steady state. To some degree the environment must be dissipative too. It must stay out of equilibrium, otherwise there is no free energy source to drive the organism. So there has to be some continual “topping-up” of food molecules in the environment to replace those that will inevitable decay or be lost. I am also confused by “non-diffusive” on line 147, and I am not sure where I am supposed to look for the “see below”.
The paper has some interesting ideas like the K concentration controlling buoyancy, and evolution of melanosomes, although these seem rather speculative. But my main point is that if life has already reached the stage where there are proteins acting as ion pumps or organelles capable of melanin synthesis, then it has already reached the point where evolution is working and anything can happen. So the problem is already solved. Potassium and melanin seem like side issues.
Round 2
Reviewer 1 Report
Comments and Suggestions for Authors
The authors have responded adequately to my criticisms. Publication is recommended without further revisions.
Reviewer 3 Report
Comments and Suggestions for Authors
Here is my suggested revision of the text, with minimal changes while preserving URLs and maintaining clarity:
The revised version of the manuscript, "On the Origin of Information Dynamics in Early Life," by Gatenby et al. provides somewhat extended discussion on how different environmental factors such as diurnal environmental variations or ion gradients may have played a role in the emergence of information behaviors in early life forms.
While I find the revised version more useful in expressing the authors' ideas, the current version still lacks clarity at many levels.
1) While information and especially its application to complex evolving systems is indeed a difficult concept, it is not an esoteric one, but rather something well quantifiable. There is some good but scarce literature available on this topic (https://onlinelibrary.wiley.com/doi/10.1002/cplx.6130010105 https://dl.acm.org/doi/10.5555/244500.244506 https://www.qeios.com/read/QNG11K.6).
2) I am still struggling with the major claims of the current manuscript. If the authors simply want to say that some additional factors have been essential for the origins of life, this is a somewhat trivial statement. For example, there is no doubt that the temperature of the early prebiotic Earth played a role in shaping life as we know it. However, one can say much more about temperature. A selection of polymers happened within a certain Goldilocks zone (a temperature range likely between 50 and 75°C). Only within this range can the polymers (proteins and RNA) fold into secondary structure, form tertiary interactions, and unfold/misfold, providing selection pressure for optimization of their sequences, structures, and functions. When the authors argue about the role of a particular factor, it is not sufficient just to say that certain properties were required. What I expect as part of their argument is why a certain factor was necessary at a given moment during the evolutionary trajectory and how a lack of a given factor may have impacted the evolutionary scenarios. In other words, I suggest that the proposed factors should be considered in more specific contexts within certain evolutionary models and scenarios.
3) There are a number of competing theories regarding the location as well as the mechanism and order of events https://www.sciencedirect.com/science/article/pii/S1674987117301305. According to one scenario, life originated in the deep sea near hydrothermal vents (see for example https://www.nature.com/articles/nrmicro1991). Within this scenario, the chemical gradients, including ionic gradients, played an essential role in the emergence of biochemical networks and early protocells. The systems emerged in non-equilibrium conditions maintained by the presence of hydrothermal vents. Yet within the framework of this scenario, it would be difficult to see the substantial role of day/night cycles or UV radiation.
4) In an alternative scenario, life emerged on the surface of the Earth at surface/water interfaces, as the early prebiotic molecules https://www.nature.com/articles/s41467-017-02639-1 gained an ability to polymerize https://www.nature.com/articles/s41467-019-11834-1 and form protocells https://www.liebertpub.com/doi/full/10.1089/ast.2019.2045 during hydration-dehydration cycles induced by thermal fluctuations at Earth's surface. Within this scenario, thermal fluctuations are the essential driving force for reactions that transform geochemistry into biochemistry. The ionic gradients would play their role somewhat later when chemical systems departed from near-equilibrium conditions within the ponds and confined themselves within protocells.
5) The manuscript contains numerous references to tRNA as if it would be the only RNA molecule required for the origins of life. The authors do not discuss rRNA and the origins of the ribosome, or mRNA and the origins of genes as the essential constituents of the translational machinery—key components of the Central Dogma of molecular biology that enables separation of genotype and phenotype and makes alterations within the genomes isolated from the selection of enzymatic functionality that these genomes encode. In addition to the above, the manuscript does not even raise the problem of the origins of RNA.
6) I still feel that the section on chirality of biomolecules is weak and incomplete. There is an enormous amount of literature on this topic but still no consensus:
https://pmc.ncbi.nlm.nih.gov/articles/PMC2857173/
https://www.sciencedirect.com/science/article/abs/pii/S0094576500000242
https://journals.plos.org/ploscompbiol/article?id=10.1371/journal.pcbi.1007592
https://www.mdpi.com/2075-1729/14/3/341
https://www.liebertpub.com/doi/10.1089/ast.2023.0007
Unless the authors can somehow contribute to the ongoing discussion by adding a substantial number of facts or arguments, I still do not see the relevance of this section.
I hope that incorporation of some of the suggestions listed above may improve the quality of the manuscript before it gets published in Life or elsewhere.
